# Red CdSe/ZnS QDs’ Intracellular Trafficking and Its Impact on Yeast Polarization and Actin Filament

**DOI:** 10.3390/cells12030484

**Published:** 2023-02-02

**Authors:** Nhi Le, Jonathan Routh, Cameron Kirk, Qihua Wu, Rishi Patel, Chloe Keyes, Kyoungtae Kim

**Affiliations:** 1Department of Biology, Missouri State University, 901 S National, Springfield, MO 65897, USA; 2Jordan Valley Innovation Center, 542 N Boonville, Springfield, MO 65806, USA

**Keywords:** quantum dots, yeast, intracellular trafficking, cytoskeleton, actin filament, polarization, growth

## Abstract

Quantum dots are nanoparticles (2–10 nm) that emit strong and tunable fluorescence. Quantum dots have been heavily used in high-demand commercialized products, research, and for medical purposes. Emerging concerns have demonstrated the negative impact of quantum dots on living cells; however, the intracellular trafficking of QDs in yeast cells and the effect of this interaction remains unclear. The primary goal of our research is to investigate the trafficking path of red cadmium selenide zinc sulfide quantum dots (CdSe/ZnS QDs) in *Saccharomyces cerevisiae* and the impact QDs have on yeast cellular dynamics. Using cells with GFP-tagged reference organelle markers and confocal microscopy, we were able to track the internalization of QDs. We found that QDs initially aggregate at the exterior of yeast cells, enter the cell using clathrin-receptor-mediated endocytosis, and distribute at the late Golgi/trans-Golgi network. We also found that the treatment of red CdSe/ZnS QDs resulted in growth rate reduction and loss of polarized growth in yeast cells. Our RNA sequence analysis revealed many altered genes. Particularly, we found an upregulation of *DID2*, which has previously been associated with cell cycle arrest when overexpressed, and a downregulation of *APS2*, a gene that codes for a subunit of AP2 protein important for the recruitment of proteins to clathrin-mediated endocytosis vesicle. Furthermore, CdSe/ZnS QDs treatment resulted in a slightly delayed endocytosis and altered the actin dynamics in yeast cells. We found that QDs caused an increased level of F-actin and a significant reduction in profilin protein expression. In addition, there was a significant elevation in the amount of coronin protein expressed, while the level of cofilin was unchanged. Altogether, this suggests that QDs favor the assembly of actin filaments. Overall, this study provides a novel toxicity mechanism of red CdSe/ZnS QDs on yeast actin dynamics and cellular processes, including endocytosis.

## 1. Introduction

Despite their tiny size, nanomaterials have caught the attention of scientists worldwide due to the broad range of their practical applications. Among many types of nanomaterials, fluorescence quantum dots (QDs) stand out as a unique candidate for many optical-based technologies in the biomedical and electronics industry. Quantum dots are nano-sized semiconductor crystals capable of emitting a wide range of size-dependent fluorescence [1,2]. Due to their unique composition and stable structure, quantum dots possess a set of desirable characteristics, such as photobleaching resistance and high quantum yield, making them highly appealing for various technologies [3,4,5]. QDs are particularly sought after for their promising biomedical applications, including cell tracking, drug delivery systems, disease detection, tumor detection, and antimicrobial remedies [6,7,8,9,10]. Furthermore, quantum dots are currently present in many high-demand commercialized products, including electronic devices, solar cells, and cosmetics [11,12,13,14].

Amid QDs’ popularity, many studies have indicated their adverse effects [15,16,17,18]. In mammalian cells, QDs can enter the cell using multiple pathways and distribute to major organelles, such as the Golgi and the lysosome [19,20]. The treatment of QDs to cells negatively impacts their mitochondrial function, increases the level of reactive species (ROS), triggers apoptosis, and reduces cell viability [21,22,23]. In vivo, QDs also have been shown to accumulate in the body of mammals and damage major organs, such as the liver, kidney, and reproductive system [21,24,25,26,27]. As QDs are also targeted for commercialized products, the toxicity of QD-containing waste products towards other biological organisms in the environment, including the fungal system, should be assessed. Yet, compared to the abundant effort in QD–mammalian interaction studies, our knowledge regarding the fungal system interaction with quantum dots and how QDs affect its biological processes are highly limited. Some current data indicated that the interaction between yeast and QDs resulted in the alteration of many genes that negatively impact the growth of yeast cells [28,29]. However, the mechanism of QD toxicity in yeast is unclear.

There are several hypotheses on the toxic mechanism mediated by quantum dots. The first main hypothesis focuses on the overall characteristics of QDs. Currently, many types of QDs are available, varying in surface ligands, shapes, and sizes—all seem to affect QDs’ cellular interactions and contribute to the toxicity of QDs [30,31,32]. In contrast, other experiments suggest that the chemical composition of QDs’ core is an important factor of their toxicity. Among the different types of QDs, cadmium-based QDs are considered to be superior in quantum yield and are suitable for many applications [33]. Unfortunately, QDs with cadmium core are thought to be more toxic compared to other core types, such as indium or carbon, since they possess cadmium ions known to be detrimental [34,35,36,37,38,39]. Thus, it is speculated that the leakage of cadmium ions during the degradation of cadmium-based QDs may be the cause of QDs’ toxicity. Supporting the hypothesis that the toxicity of QDs stems from the leakage of core materials, many studies have demonstrated the effectiveness of an external protective shell, such as ZnS shell, in decreasing QDs’ toxicity [40,41,42,43]. In contrast, a study highlighted the difference in cellular response when yeast cells are exposed to cadmium ions versus when they are exposed to whole quantum dots [44,45], implying that core leakage is not the sole factor responsible for QDs’ toxicity. Recently, some studies have suggested that QDs could be readily internalized by fungal cells despite the rigid cell wall [46,47], giving rise to a new hypothesis that QDs’ toxicity could have resulted from the internalization of QDs in fungal cells.

As cadmium-based QDs possess the potential for many applications, it is important to assess their potential toxicity to ensure the safety of our environment and the different living systems. In this study, we chose the budding yeast *Saccharomyces cerevisiae* as our model organism due to its abundance in nature and industrial importance. We aim to investigate the interaction between red cadmium selenide zinc sulfide quantum dots (CdSe/ZnS QDs), a type of cadmium-based core-shelled structure quantum dots with negatively charged carboxylic ligand attached, and *Saccharomyces cerevisiae*. Furthermore, we tracked QDs’ intracellular trafficking in yeast and revealed the potential mechanism for actin dynamic alteration in yeast upon QD exposure.

## 2. Materials and Methods

### 2.1. Yeast Strains

The strain expressing green-fluorescence-tagged cofilin protein (GFP-Cof1) was generated by acquiring plasmids from Addgene (*pRS316-GFP-COF1*). The plasmid was isolated by using the QIAperp Spin Miniprep Kit CAT. NO. 27104 (QIAgen, Hilden, Germany) following the manufacturer’s protocol. The concentration of the isolated DNA contents was measured using Implen Nanophotometer (Implen, Munchen, Germany). We ensured the presence of the isolated DNA by running agarose gel electrophoresis at 140 volts, 3.00 A for 30 min. Afterward, we inserted the isolated DNA into 3 mL of mid-log growth-phase yeast culture by using a one-step transformation protocol [48]. The mixture includes 5 μL of DNA (174 ng/μL), 5 μL of 2 mg/mL denatured salmon sperm DNA, and 90 μL of filter-sterilized one-step transformation buffer (0.2 M lithium acetate, 40% PEG3400, 100 mM DTT, and DI water). The mixture was vortexed and incubated in a 45 °C water bath for 45 min. Afterward, the mixture was bead-plated on a nutrient-lacking uracil plate (SD-URA) and incubated for three days. Three colonies were picked and screened by using the Olympus IX81 ZCD2 fluorescence microscope. Positive colonies were saved with 20% glycerol at −80 °C for long-term storage.

For a new strain expressing green-fluorescence-tagged actin-binding protein 140 (Abp140-GFP), we amplified the GFP sequence from pFA6a-GFP(S65T)-His3MX6 vector for tagging the GFP sequence at the 3′ end of ABP140 gene in the chromosome by following the polymerase chain reaction (PCR) protocol previously described [49]. The primers used for the PCR reaction are as follows: the forward primer sequence was GTACCGCTGCTGGGTACAA-GCTGTGTTTGACGTTCCTCAACGGATCCCCGGGTTAATTAA and the reverse primer sequence was TTTATGATGAGAGAGGAGGTGGTACTT-GTCTCAGAACTTCGAATTCGAGCTCGTTTAAAC.

We then amplified the desired DNA using PCR, followed by the transformation of competent cells using the Frozen-EZ Yeast Transformation II Kit Cat T2001 (Zymo Research, Irvine, CA, USA). We bead-plated the cells on an agar plate and allowed them to grow. We chose a few colonies and visually confirmed the strain under a confocal microscope described below. We selected and saved the colonies with glycerol at −80 °C for long-term storage. Some of the strains, including wildtype strain S288C, strain expressing actin-binding protein 1 tagged with green fluorescence (Abp1-GFP), strain expressing GFP-Snc1, strain expressing GFP-2PH, and strain expressing vacuolar protein sorting 10 tagged with green fluorescence protein (Vps10-GFP), were created by prior lab members and saved with glycerol at −80 °C (Table 1).

### 2.2. Yeast Culturing

Before each experiment, we streaked yeast strains (Table 1) saved at −80 °C on appropriate selective agar plates, including yeast peptone dextrose (YPD), nutrient-lacking histidine or SD-HIS, nutrient-lacking leucine or SD-LEU, and nutrient-lacking uracil or SD-URA (Table 1), and incubated them for a few days in a 30 °C incubator. Then, an isolated colony was selected, moved to 3 mL of corresponding liquid media, and grown for 24 h at 30 ℃ to create a fresh stock. We transferred cells to 500 μL of fresh media to make samples with an optical density of 0.1 at 600 nm. These samples were treated with QDs that had been water sonicated for 30 s.

### 2.3. CdSe/ZnS QDs Characteristics

Cadmium selenide/zinc sulfide quantum dots were obtained from NN-Lab (AR, catalog number: CZW-R-5) with an emission peak of 620–635 nm, containing carboxylic acid ligand (<1% organic impurities, not including ligands) suspended in water (1 mg/1 mL).

Zhang et al. measured CZW-R-5 (red CdSe/ZnS QDs) using the dynamic light scattering (DLS) technique and revealed the major fraction of CZW-R-5 displayed a hydrodynamic diameter of 20 nm [50]. Using scanning electron microscopy, they showed that CZW-R-5 were around 5–10 nm when air-dried [50]. These numbers are consistent with the manufacturer’s information.

### 2.4. Dynamic Light Scattering

For dynamic light scattering (DLS) experiments, QDs (50 μg/mL) were dispensed in 9 mL of water, SD-HIS, or YPD. After 6 h of incubation, the samples were delivered to Jodan Valley Innovation Center (JVIC, MO, USA) to measure the hydrodynamic size of QDs using the Colloid Metrix NANO-flex^®^ II (Meerbusch, Germany) with laser wavelength at 632 nm and a scattering angle of 180°.

### 2.5. Inductive Coupled Plasma Atomic Emission Spectroscopy

For the inductively coupled plasma atomic emission spectroscopy (ICP-OES) experiment, 50 μL (50 μg/mL) of QDs was added to water with pH ranging from 1 to 7 or culture media. The samples were incubated stationarily for 24 h at room temperature. Afterward, the samples were spun at 1500 rpm, and the supernatant was collected. The samples were delivered to JVIC. The amount of cadmium (Cd) in the water samples was analyzed by inductively coupled plasma optical emission spectroscopy (ICP-OES, iCAP7400 Duo, Thermo Scientific, Waltham, MA, USA).

Nitric acid (68–70% in water, Acros Organics) was of ACS grade and purchased from Fisher Scientific (Waltham, MA, USA). Deionized (DI) water was generated by Milli-Q^®^ IQ 7000 Ultrapure lab water system (MilliporeSigma, Burlington, MA, USA). Stock solutions of calibration standards (1000 ppm, 23 elements) and internal standards (IS, 100 ppm Yttrium) were purchased from Inorganic Ventures (Christiansburg, VA, USA). All quality control (QC) samples and calibration standards were prepared by appropriate dilution with 1% HNO_3_, and the diluted acid (1% HNO_3_) was used as both rinse solution and reagent blank.

Water samples were transferred into 15 mL Falcon tubes and directly tested by ICP-OES within 48 h. A total of 19 elements (including Cd) were analyzed by ICP-OES according to EPA Method 200.7. [48] with minor modifications. The accuracy of the test method was monitored by analyzing QC standards for every 5 samples.

### 2.6. Confocal Microscope

For all fluorescence microscopy images and videos, we used the Olympus IX-81 inverted microscope (Olympus Corporation, Tokyo, Japan) equipped with a Yokogawa CSU-X1 spinning confocal box (Yokogawa, Sugar Land, TX, USA) and ImagEM camera (Hamamatsu Photonics, Shizuoka, Japan). For the subcellular colocalization of QDs, we used both green (561 nm) and blue (488 nm) lasers to record the images. For other GFP detection experiments, the blue laser (488 nm) was used. The 100× objective and the SDC single filter were selected for the focus window. Approximately 2 μL of cells ws released on a glass slide and covered with a coverslip, and one drop of oil was added to enhance visualization. Afterward, the flat images, 3D images, and videos were taken.

### 2.7. QDs Interaction and Subcellular Localization Detection by Confocal Microscopy

We cultured strains expressing GFP-2PH, Abp1-GFP, FAPPI-GFP, or Vps10-GFP (Table 1) as stated in Section 2.2. A 500 μL sample containing yeast cells with 0.1 optical density was treated with 4 μg/mL of red CdSe/ZnS QDs and incubated over time (0 h, 3 h, 6 h, and 24 h). After the designated incubation time, we visualized cells using the confocal microscope with a 488 nm laser channel and a 561 nm laser channel. Two-dimensional flat images, 3D images, and a 1-min 2D timelapse video (2 frames/s) were taken. Three images from each sample were selected (N = 3) to be analyzed for colocalization using the Pearson’s coefficient function in the ImageJ program plugin JACoP, as described by Zhang et al. 2022 [50]. The process was repeated in a triplicate manner. Data were then graphed using Prism GraphPad 9.

### 2.8. GFP-Snc1 Polarization

We cultured GFP-Snc1 as stated in Section 2.2. Then, we prepared nontreated control (NTC) samples and samples treated with 4 μg/mL, 12 μg/mL, 25 μg/mL, or 50 μg/mL of CdSe/ZnS QDs. Then, we incubated them over time (1 h, 3 h, 6 h, and 24 h). Next, we visualized the cells using the confocal microscope. For each sample, a total of 30 cells (N = 30) were assessed for polarity and used to calculate the percentage of polarized cells. Cells with GFP-Snc1 concentrated at the bud area are considered to be polarized. Cells with GFP-Snc1 distribution to the neck and mother site are considered to be depolarized. We performed additional runs to triplicate the experiment. The percentages of polarized cells were entered and then graphed using Prism GraphPad 9.

### 2.9. Viability Assay

Samples of 500 μL of Abp1-GFP cells grown in SD-HIS with an optical density of 0.1 were treated with different concentrations of CdSe/ZnS QDs (4 μg/mL, 12 μg/mL, 25 μg/mL, and 50 μg/mL) and incubated for 6 h and 24 h. After the indicated time, we recorded the optical density of the samples and graphed the data using Prism GraphPad 9.

### 2.10. Kinetic Growth Assay

A fresh stock of Abp1-GFP strain (Table 1) was diluted in SD-HIS media to an optical density of 0.1 at 600 nm. Next, we added 100 μL of the diluted yeast to each well in a 96-well round-bottom assay plate (Corning Incorporated, Kennebunk, ME, USA). Various concentrations of QDs (0 μg/mL, 4 μg/mL, 12 μg/mL, 25 μg/mL, or 50 μg/mL) were added to the wells. The plate was inserted into the BioTEK ELx808 Incubating Absorbance Plate Reader ( BioTek, Winooski, VT, USA). We measured and recorded the optical density at 594 nm every 30 min over the course of 24 h using a preset protocol on Gen5 Microplate Reader and Imager Software (BioTek, Winooski, VT, USA). The experiment was quintuplicated. We graphed the data using Prism GraphPad 9.

### 2.11. RNA Sequencing

Yeast cultures of 25 mL with an optical density of 0.1 were created and treated with 25 μg/mL of red CdSe/ZnS and incubated for 6 h. Afterward, we followed the protocol of the Invitrogen RiboPure-Yeast Kit (Thermo Fisher Scientific, Vilnius, Lithuania) to extract total RNA. The total RNA concentration and quality were measured using an Agilent Bioanalyzer. Next, we sent the samples (1–2 μg total for each sample) to Novogene Lab in California for next-gen sequencing. The raw sequencing data (Fastq files) received from Novogene Lab were downloaded and concatenated using the Galaxy website (usegalaxy.org). The concatenated files were imported into the CLC Genomic Lab Bench 22.0.0 program (QIAgen, Hilden, Germany). The files were then checked for quality control and sequence lengths below 15 and greater than 1000 were trimmed. The trimmed files were subjected to another quality control check. The sequence was then analyzed with the standard *Saccharomyces cerevisiae* yeast reference genome. Up- and downregulated genes with a fold change of 1.25 and a *p*-value < 0.05 were selected for analysis. These genes were then categorized by protein class and biological processes using Panther Gene Ontology (pantherdb.org).

### 2.12. Actin Filament and Actin Binding Protein Assay

Samples of yeast with an optical density of 0.1 at 600 nm were treated with 4 μg/mL, 12 μg/mL, 25 μg/mL, or 50 μg/mL of red CdSe/ZnS QDs and grown at 30 ℃ for 6 h. After incubation time, we fixed the samples with 3.7 % formaldehyde, sonicated them on ice using the Branson Sonifier probe (Branson Ultrasonics, Brookfield, CT, USA) for 1 min, and washed with cold 1 × PBS containing 1% triton-X-100. Next, we resuspend the samples with cold 1 × PBS and collected the pellet by centrifugation. A 1/30 volume of cold 1 × PBS was added and 10 μL of cells were mixed with 3 μL of 14 μM Actistain 488 dye for 20 min with gentle agitation. The cells were then washed twice with cold 1 × PBS and visualized using the confocal microscope with a green laser (excitation wavelength of 561 nm). For each treatment group, 24 cells were analyzed (N = 24) and used to calculate the percentage of cells with abnormal actin filament, and the experiments were triplicated to obtain the mean and the standard deviation. Cells with smooth, continuous, and straight cables are considered normal. Cells with noncontinuous or digited, curved, or thicker cables are considered to be abnormal.

For an actin filament recovery experiment, samples of yeast with an optical density of 0.1 at 600 nm were treated with 4 μg/mL, 12 μg/mL, 25 μg/mL, or 50 μg/mL of red CdSe/ZnS QDs and grown at 30 ℃ for 6 h. The cells were then washed, resuspended with fresh QDs-free media (SD-HIS), and incubated for an additional 3 h. Afterward, the same process for actin staining and data analysis was followed.

### 2.13. Flow Cytometry

Strains expressing Crn1-GFP, Pfy1-GFP, or GFP-Cof1 (Table 1) were grown to acquire fresh stock. We transferred cells to 500 μL of fresh media to make samples with an optical density of 0.1. These samples were treated with 50 μg/mL of QDs that had been water sonicated for 30 s. The samples were incubated for 6 h, after which we washed the samples by centrifugation. The fluorescence intensity of the cells was measured by the Attune NxT flow cytometer (Thermo Fisher Scientific, Waltham, MA, USA) using BL1-A channel (488 nm). Ten samples were assessed per treatment group and the median fluorescence intensity was recorded. The experiments were duplicated. Finally, we graphed the result using the unpaired, parametric T-test function of the GraphPad Prism 9 software.

To measure the level of F-actin, a wildtype S288c strain with no tagged fluorescence protein was grown overnight to acquire a fresh stock. The samples were generated and treated as mentioned above and incubated for 6 h. Then, the actin staining protocol described in Section 2.7 was followed. The fluorescence intensity of stained actin cables was measured by the Attune NxT flow cytometer using a BL1-A channel (488 nm). Ten samples were assessed per treatment group and the median fluorescence intensity was recorded. The experiments were duplicated. Finally, we graphed the result using the unpaired, parametric T-test function of the GraphPad Prism 9 software.

### 2.14. Statistical Analysis

All graphs were made by GraphPad Prism 9.0 using one-way ANOVA and Dunnett’s multiple comparisons or unpaired, parametric *t*-test. All experiments were triplicated unless otherwise noted, and variations were shown on graphs as error bars. Statistically significant data on the graph is represented as * *p* < 0.05, ** *p* < 0.01, *** *p* < 0.001, and **** *p* < 0.0001.

## 3. Results

### 3.1. QDs Agglomeration Assay

A previous study suggested that the overall characteristics of QDs, such as ligand [51] of QDs, are the leading cause for their toxicity, while other studies inferred that the leakage of core ions during material degradation may be responsible for the toxicity of QDs [34,35]. Therefore, QDs’ structural stability undoubtedly influences their interactions with biological systems. It has been reported that QDs agglomerate in some biological media [52]. This formation of clusters is thought to have been caused by the loss of QD surface components. Thus, we decided to test the stability of QDs in different pH and different selective media by assessing the agglomeration of QDs.

Through DLS, we saw that the average size of QDs in water at normal pH was relatively small (average diameter of 30.23 nm, Table 2), indicating minimal QD agglomeration in this condition. Additionally, QDs retained a negative zeta potential in water, hinting at an intact negatively charged carboxylic ligand layer. Both characteristics show that the structure of QDs is stable in water. In contrast, QDs suspended in nutrient-lacking histidine media (SD-His) were about 47 times larger compared to QDs suspended in water and had a mildly positive zeta potential (Table 2). These observations revealed that, in SD-His media (pH of 4.86), QDs may have significantly agglomerated into large clusters. Lastly, in yeast YPD media (pH 5.75), QDs were around eight times larger than QDs in water and showed a relatively neutral zeta potential (Table 2).

As the tested solutions had different pH, we decided to investigate if pH is the cause for QDs’ instability. Quantum dots were added to water with a pH of 1 to 7 and incubated for 24 h. Afterward, we spun the samples to eliminate the largely agglomerated QD clusters and measured the cadmium content that remained in the supernatant. We found that the amount of cadmium in samples with water of pH 2–7 was consistent with the control and only noticeably dropped in the pH 1 sample (Table 3). These data hinted that QDs are relatively stable in pH above 2 and that pH is not the main reason for the agglomeration of QDs.

To confirm the hypothesis that pH is not the main factor for the clustering of QDs, we tested QDs in SD-HIS and YPD media in their natural pH (4.86 and 5.57, respectively), as well as neutral pH 7. The results showed, beside a moderate drop in Cd^2+^ ion content in SD-HIS media at pH 4.86, there was not a major difference between the media with different pHs (Table 4). However, when compared to the data in Table 3, the Cd^2+^ ion content in media was only about half of the Cd ion content in water with a pH above 2. Thus, this indicates that the components of SD-HIS and YPD media are responsible for the agglomeration of QDs rather than pH.

### 3.2. QDs Interaction and Subcellular Trafficking in Saccharomyces cerevisiae

#### 3.2.1. QDs Initial Site of Attachment

The interaction between red CdSe/ZnS QDs and yeast cells was visualized with the confocal microscope. Initially, QDs were found attaching to the outer surface of yeast cells (Figure 1A). The attachment of QDs on yeast began momentarily after the QD treatment and was more frequently found in the mother cell region rather than at the bud or neck (Figure 1B). Although there was a decrease in the number of cells with QDs attached on the surface after 8 h (data not shown), the cells that do have associated QDs still showed a preference for the mother region.

#### 3.2.2. QDs’ Subcellular Trafficking

Due to the lack of information regarding the intracellular trafficking of quantum dots in yeast, we tracked QDs’ fate throughout 24 h. With the plasma membrane reference marker GFP-2PH [53], we observed a rim of associated QDs on the surface of yeast cells (Figure 2A). Although minutely shown to overlap with the plasma membrane shortly after the QD treatment, according to a Pearson’s correlation test, QDs did not appear to be completely colocalized with the plasma membrane reference marker until around 3 h after QDs were added (Figure 2A,B). It was also found that the amount of QDs overlapped with the plasma membrane reference marker is at its highest at around 6 h, as indicated by the yellow appearance of the plasma membrane and the high Pearson’s correlation coefficient (Figure 2A,B). At 24 h, QDs were no longer found at the plasma membrane (Figure 2A,B), likely due to the lack of QD supply, as QDs were found to aggregate in media over time (data not shown). In previous studies, QDs were found to enter HeLa cells mainly via endocytosis pathways [19]. To examine whether the receptor-mediated endocytic (RME) route is used in yeast for the QD internalization, we used a reference marker that marks endocytosis patches, Abp1-GFP. The result revealed that QDs do not arrive at the RME site until about 3–6 h after treatment (Figure 2A,C).

Next, we assessed whether QDs colocalize to the late Golgi/*trans*-Golgi network (TGN), and we were also able to detect the presence of QDs 3–6 h after treatment, as suggested by the confocal microscopy image in Figure 3A,B. This is consistent with a recent finding suggesting that, in yeast, the TGN also serves as the early endosome and recycling endosome [54]. It is well known that an endosome maturates into the late endosome before fusion with the lysosome. Therefore, we investigated whether QDs colocalize to Vps10-GFP, a late endosome reference marker [55], to assess whether QDs are targeted to the lysosome for degradation. We observed a very low level of partial colocalization with Vps10-GFP, which was confirmed by the low Pearson’s coefficient values (Figure 3A,C). This suggests that QDs may not primarily localize in the endosome.

### 3.3. Pinocytosis Colocalization Assay

As yeast uptakes materials using multiple pathways, we decided to investigate if QDs also enter yeast through pinocytosis. Our confocal microscopic images revealed that QDs partially colocalize with the pinocytosis reference dye (Fm1-43, Figure 4A). When conducting a Pearson’s coefficient correlation analysis, we found low Pearson’s coefficient values (Figure 4B). However, the Pearson’s correlation coefficient values of the treated groups showed an increasing trend over time (Figure 4B), indicating that pinocytosis may play a minor role in QD trafficking.

### 3.4. Growth Assay

After observing the internalization of QDs, we were interested in studying the effect of QD treatment on yeast cells. To fully examine QDs’ impact on yeast growth, the optical density of yeast cultures was obtained (Figure 5). At 6 h post-QD treatment, yeast growth was significantly reduced, even in the lowest concentration of 4 μg/mL. Furthermore, the reduction in yeast growth correlated to the increase in QD concentration (Figure 5A). However, after 24 h of QD exposure, there was no significant difference in the optical densities between the groups (Figure 5B). This is likely because, 24 h after QD treatment, the fraction of QD-resistant cells might have grown over time, reaching the steady state. Consistently, our kinetic growth curve also showed a slight delay in the lag phase in groups treated with quantum dots (Figure 5C,D). Similar to our observation, Horstmann et al. recently reported that the lag phase of yeast cells treated with yellow and green CdSe/ZnS QDs is prolonged compared to nontreated cells [28]. Thus, the prolonged lag phase could be one of the factors that caused delayed growth rate in QD-treated yeast samples.

### 3.5. GFP-Snc1 Distribution Altered by QDs Treatment

To find the cause of delayed growth caused by red CdSe/ZnS QDs, we examined the level of GFP-Snc1 polarization in yeast cells by using a strain expressing GFP-Snc1. Polarized growth is an important function of cells, where growth is asymmetrically focused on the bud [56], and, therefore, the loss of polarization could give an insight into why growth in yeast cells is delayed. We treated yeast cells expressing GFP-Snc1 with an increased dosage of QDs (4 μg/mL, 12 μg/mL, 25 μg/mL, and 50 μg/mL) and visualized GFP-Snc1 distribution on the plasma membrane (Figure 6A). We found that QDs did not affect the polarization of yeast cells in the first 3 h of treatment (Figure 6B,C). After 6–24 h of QD treatment, while lower treatment concentrations showed no significant difference from the control group, higher concentrations of QD treatment, 25 μg/mL and 50 μg/mL, caused a reduction in the number of polarized cells (Figure 6D,E). This suggests that the loss of polarized growth could have been a factor that led to the slowed growth in QD-treated cells. More examples are shown in Appendix A.

### 3.6. Transcriptomic Analysis

To fully understand the impact of QDs on yeast cells, we performed an RNA-sequence experiment. Yeast cells were treated with 25 μg/mL of red CdSe/ZnS QDs, the minimum amount that affected GFP-Snc1 polarization. Total RNA of the samples with and without QD treatment was extracted, followed by cDNA sequencing and quantification. All groups were triplicated. The samples were then sent to Novogene Lab, located in California, to sequence cDNA, and the sequenced data were processed by the CLC workbench 22.0.0 software. A total of 215,713,965 (97.71%) reads were mapped for the control groups and 227,006,038 (97.54%) reads were mapped for the treated groups. Differentially expressed genes (DEGs) with a fold change greater than 1.25 or smaller than –1.25, with a *p*-value less than 0.05, were selected and sorted into their corresponding protein class and associated biological processes using the Panther Gene Ontology website. Results showed that QD treatment caused the downregulation of 271 genes and the upregulation of 47 genes (Figure 7A). From these, many downregulated genes were found to encode for proteins that participate in the translation, metabolism, cell wall organization, cytoskeleton, and membrane trafficking (Figure 7B). These genes are also associated with cellular processes, metabolic processes, and cellular organization (Figure 7D). In comparison, fewer genes were upregulated. These genes are metabolic and membrane trafficking regulating genes that are involved in cellular and metabolic processes (Figure 7C,E, Table 5). A model of important up- and downregulated genes has been provided (Figure 8).

Among the upregulated genes, *DID2* is the gene with the highest fold change (82.3, *p*-value is 0.025) (Table 5). Did2 is a key regulator of the endosomal sorting complex (ESCRT), and it plays an essential role in endosomal transport co-ordination [57]. The overexpression of *DID2* was previously associated with a cell cycle arrest in the G2 and M phase [58]. Thus, the upregulation of *DID2* might have induced a cell cycle arrest in our QD-treated cell culture, which may explain the observed inhibitory effect on the growth of treated cells (Figure 5). Other upregulated genes involved in vesicular trafficking include *COS10*, which belongs to the COS gene family involved in sorting nonubiquitinated cargos to the multivesicular bodies [59,60]; *ECM21* involved in the endocytosis process; and *TRS85*, a subunit of transport protein particle complex (TRAPP) [61] that is important for the structure of pre-autophagosome [62]. The overexpression of these genes indicates possible defects in cellular vesicular trafficking processes upon CdSe/ZnS QDs treatment.

The RNA-seq data revealed many downregulated genes in response to CdSe/ZnS QDs treatment (Table 6). Adaptor protein 2 (*APS2*) is important in initiating the assembly of clathrin-coated pit proteins when activated [57]. It was also reported that *APS2* plays an essential role in the polarization of certain proteins on the plasma membrane [63]. The downregulation of this gene (Table 6) hinted that the endocytosis pathway of yeast might be impacted by the presence of QDs. Therefore, we hypothesized that the endocytosis process of QD-treated cells would be inefficient compared to nontreated cells, leading to inadequate nutrient uptake and resulting in a slower growth rate.

### 3.7. Endocytosis-Associated Protein Dynamic Defects

Due to the downregulation of the AP2 component found in our RNA sequence data, we decided to investigate the efficiency of endocytosis by analyzing the dynamics of Abp1-GFP, a reference marker for receptor-mediated or clathrin-mediated endocytosis [64]. Results showed that the lifespan of Abp1-GFP at endocytic patches at the plasma membrane of yeast cells treated with 25 μg/mL of CdSe/ZnS QDs was 2–3 s longer compared to that of nontreated cells, which hinted inefficient endocytosis (Figure 9A,B). This observation is consistent with our previous prediction based on the downregulation of *APS2* genes. Furthermore, we also observed a 3 s delay in the disassociation of Abp1-GFP from the post-endocytic vesicle after membrane detachment (Figure 9C). The data suggest that QDs have an adverse effect on the invagination of endocytic vesicles from the plasma membrane, thus negatively impacting yeast’s ability to take up nutrients and leading to slow growth. Furthermore, our kymograph analysis shows that the endocytic patches in treated groups stay longer at the membrane.

### 3.8. Actin Filament Assay

#### 3.8.1. Actin Filament Abnormality

It is well known that F-actin-carrying structures, including actin patches and actin cables, play an essential role in the regulation of endocytosis [65,66,67,68,69]. The actin filament also is a key factor in regulating and maintaining the polarization in yeast asymmetric growth [70,71,72,73]. For this reason, we used Actistain 488 dye to investigate the impact of QDs on yeast F-actin. For this experiment, we treated cells with QDs for 6 h, fixed them with formaldehyde, and stained the cells with phalloidin-conjugated Actistain 488. The result showed that QD concentration of 12 μg/mL and above caused an abnormal appearance in the actin cable (Figure 10A,B). Compared to the smooth cables in the control group, cells treated with high concentrations of QDs were thicker and showed a branch-like appearance (Figure 10A). The altered morphology of the actin cable hinted at an unbalanced actin dynamic by favoring more assembly of F-actin, leading to a slow turnover rate for actin disassembly. The change in the turnover rate may play an essential role in the observed reduction in yeast growth by effecting actin-mediated endocytosis.

The actin cables were observably brighter in groups treated with high concentrations (Figure 10A). Thus, we used the line function provided by Slidebook v6 to examine the intensity of actin filaments of the control and the treated cells. There was a moderate increase in actin intensity for the QD-treated cells compared to the control cells, suggesting that there is a slight increase in the level of the F-actin filament.

To confirm this result, we visualized Abp140-GFP, which is a cable-binding protein to assess the in vivo effect of QDs on actin cable. The result showed that, at 6 h after QD exposure, actin cable formed network-like branches compared to the control (Figure 11A,B), supporting the result found in in vitro actin staining experiment.

#### 3.8.2. Actin-Associated Proteins

It is well known that the process of actin nucleation and depolymerization is modulated by a number of proteins, including cofilin [74,75], coronin [76], and profilin [77]. In normal conditions, these actin-associated proteins work together to maintain a balanced actin dynamic. To search for the mechanism of thickened actin cable, we measured the level of actin dynamic regulatory proteins tagged with GFP. We also looked at the level of filamentous actin to confirm whether there is more F-actin. We detected an increase in the level of F-actin (Figure 12A), which suggests that QDs may promote the polymerization of the actin filament. We found a decreased level of profilin (Pfy1-GFP; Figure 12B), a protein that is often associated with actin filament nucleation. The results seem contradictory; however, it has been reported that profilin also acts as a temporary sequester of monomeric G-actin to prevent spontaneous polymerization and binds to the barbed end to prevent branching of the actin filament [78]. Thus, a low level of profilin may lead to spontaneous actin filament formation, causing a branched filament network that increases the amount of F-actin.

Our data also revealed that the level of cofilin (GFP-Cof1) is unchanged (Figure 12C). Cofilin is an important protein whose function is to sever actin filaments and promote depolymerization. Thus, this finding supports the idea that QD treatment does not affect cofilin and its function in the actin dynamic. Furthermore, our data revealed an increase in coronin level (Crn1-GFP) (Figure 12D). Coronin normally binds to cofilin to enhance cofilin’s severing activity; however, coronin also protects filaments rich in ATP from premature depolymerization [79]. Overexpression of coronin was also found to cause branching and nucleation of actin filament [80]. Collectively, our data show that QD exposure led to a shift toward actin nucleation and bundling.

#### 3.8.3. Actin Filament Recovery Assay

Impacts resulting from short QD exposure have been found to be repairable by cells [81]. As an attempt to investigate the reversibility of QD-induced abnormality in actin cables, we performed a recovery assay. We incubated cells with QDs for 6 h; then, we washed and incubated cells in fresh, QD-free media for about 3 h. Afterward, we used the same procedure mentioned above and assessed the appearance of the F-actin cable. We found that actin cables of cells previously treated with QDs concentration 25 μg/mL and lower were not significantly different from the control group (Figure 13A,B). A high percentage of cells that were previously treated with 50 μg/mL still had thick, branched cables (Figure 10B); however, the cables of these cells appear to be smoother compared to prewashed cells. This suggests that the effect of short-term QD exposure is not permanent.

## 4. Discussion

Recently, the high demand for nanomaterials’ application has resulted in concerns about the impact of nanomaterial waste on humans and the environment [82,83]. As yeast is important to both the environment and human daily life, its interaction with QDs should be thoroughly researched. To the best of our knowledge, we are the first to investigate the trafficking of QDs in yeast, giving hints and insights into the intracellular interaction with QDs. By studying QD-induced transcriptomic change and protein expression change, we revealed the mechanism of QDs’ impact on the yeast actin cytoskeleton. Our data provide evidence that QD-induced actin polymerization is opposite from the impact previously observed in cadmium ions, which favors depolymerization of F-actin. Thus, collectively, our research supports that QDs’ toxicity results from QDs themselves instead of ion leakage as previously speculated.

### 4.1. QDs Agglomeration in Media

The size of QDs in culturing media is relevant when it comes to QDs’ toxicity, as various studies discovered that certain sizes of QDs can internalize into yeast and fungal cells [84]. However, it was shown that QDs can aggregate in certain biological media [52]. Furthermore, the agglomeration of QDs can indicate the instability of QDs’ structure, which is an important implication when discussing ion leakage as a possible toxic mechanism of QDs. Thus, we wanted to investigate the size and stability of QDs in our media. We found that the agglomeration of QDs in the media resulted from a factor different from pH. A recent study showed that the presence of monovalent calcium ions (NaCl) and divalent calcium ions (CaCl_2_) could promote aggregation of QDs. In particular, the presence of divalent calcium ions caused the formation of larger sizes of QD clusters by acting as a bridge to link the carboxylic ligands on QDs together [85]. Out of the 19 measured elements in our ICP-OES experiment, we discovered a high level of manganese (Mn), magnesium (Mg), and calcium (Ca) in our SD-HIS media compared to YPD. Thus, we suspected that the presence of these electrolytes could have acted as a bridge to link the carboxylic ligand of QDs together. This discovery influenced the experimental design of our subsequent experiments. In trafficking assays, we cultured yeast in YPD to minimize the agglomeration of QDs, which could potentially lead to sequestration of the QD supply in the media. For the other experiments that require the accuracy of GFP-fluorescence detection, such as measuring the expression of Crn1-GFP and Pfy1-GFP, we used selective media to acquire the most accurate results for the level of an expressed protein. Our data suggested that, after 24 h of incubation in culturing media, around 50 percent of QDs did not agglomerate and remained in the supernatant and were not removed by centrifugation (Table 2 and Table 3). These non-agglomerated QDs remained small in size and could have directly interacted with yeast by entering into yeast cells. It is worth noting that, in the ICP-OES experiment, QDs were suspended and incubated in culture media for 24 h, while most of the impacts shown in our study happened at around 6 h. It is possible that, at 6 h, QDs have yet to severely agglomerate and, thus, more small QDs were available for direct interaction.

### 4.2. Trafficking of QDs in Saccharomyces cerevisiae

The intracellular trafficking of nanomaterials determines its interactions with subcellular components and implicates their toxicity mechanism. However, currently, the exact trafficking route of QDs in yeast cells is unknown. Thus, we investigated the intracellular trafficking of red CdSe/ZnS QDs in yeast cells. We found that QDs internalize into yeast cells mainly using receptor-mediated endocytosis and travel from the endocytic vesicle to the Golgi. This finding is consistent with the reported QD trafficking route in mammalian cells [19]. In yeast, nanomaterials targeted to the degradation pathway are known to travel from the TGN directly to the late endosome [19]. However, our data suggest that the lysosome (vacuole) may not be a primary destination for QD trafficking in yeast. As a number of genes implicated in the Golgi–vacuole pathway were altered upon QD treatment (Table 5 and Figure 8), this phenomenon could have resulted from a disrupted Golgi-to-vacuole trafficking pathway (Figure 14). Particularly, we found overexpression of *DID2*, which is a regulator of the ESCRT complex [86]. *DID2* plays a key role in sorting and maintaining a balance in endosomal transport [87]. Thus, an altered expression of *DID2* may cause dysregulation of this intracellular trafficking route, which is important for the transportation of a number of important proteins [88,89]. Furthermore, Golgi-derived vesicles participate in transporting hydrolases to the vacuole required for autophagy and degradation [90]. Therefore, the impaired Golgi–vacuole trafficking may participate in the delayed growth of yeast due to inefficiency in the degradation of waste materials.

### 4.3. QDs Altered Actin Dynamic Regulatory Proteins

When further investigating the mechanism of QDs’ toxicity on yeast, we discovered that QDs cause inefficient endocytosis. The endocytosis inefficiency of treated cells likely resulted from the downregulation of *APS2*, which is a component of the AP2 adaptor. AP2 is essential in regulating the recruitment of clathrin to the endocytic pit [57]. The alteration of *APS2* expression may cause a delay in endocytic pit maturation and result in a slow endocytosis process. Furthermore, a recent study suggests that AP2 also plays an essential role in the polarized distribution of membrane proteins [63]. When assessing the distribution of GFP-Snc1 on the plasma membrane, we saw that QDs caused severe depolarization of GFP-Snc1. As polarized growth is essential in yeast cells [56,71], the loss of polarized growth could also have contributed to the delayed growth rate (Figure 5), thus providing another additive mechanism in QD toxicity towards yeast cells. An essential component in regulating the internalization of receptor-mediated endocytosis and polarized growth in yeast is actin cytoskeleton [70]. Therefore, we decided to investigate yeast actin cytoskeleton upon QD treatment. We found that QD treatment caused an elevation of the F-actin level. Accompanied by this, we saw a decrease in the level of profilin protein, which plays a role in sequestering monomeric G-actin to prevent spontaneous nucleation [78]. The lack of profilin protein may have resulted in random nucleation from ATP-rich G-actin, which led to more F-actin. Additionally, we found that QDs cause an elevation in coronin protein expression. The overexpression of coronin has previously been associated with actin bundling and crosslinking [80,91]. Therefore, this might explain our observation of brighter, thicker, and more bundled cables. Although the impact of QDs on yeast actin has not been investigated, a study testing the impact of cadmium ions on the actin cytoskeleton has found that cadmium ion promotes disassembly of actin filaments. Specifically, cadmium ions caused downregulation of profilin and increased levels of cofilin and coronin protein [92]. Furthermore, in rat renal mesangial cells, cadmium ions were also found to cause dimerization of CAP1 protein and aid its function in enhancing cofilin-mediated actin severing [93]. These alterations signified that cadmium ions by themselves favor the depolymerization of the actin filament, which is the opposite of our findings in cadmium-based quantum dots (Figure 12. Therefore, our finding supports that the toxicity of CdSe/ZnS QDs is due to QDs’ structure as a whole and not due to cadmium leakage. Given that actin plays a role in endocytosis [65], the alteration of the actin dynamics could also contribute to the hampered endocytic vesicle’s movement. Overall, alteration of these components may have impacted yeast’s ability to uptake nutrients, resulting in the observed slowed growth rate of yeast cells.

### 4.4. QDs’ Effects on the Transcription Mechanism

It is worth noting that our RNA-sequencing results hinted that our red CdSe/ZnS QDs had less impact on yeast transcriptome compared to green CdSe/ZnS QDs, which caused alterations to thousands of genes emphasized in metabolic processes and cellular organizations [28,29]. This is consistent with recent findings showing QDs’ size contributes to QD toxicity, where larger QDs are considered less toxic compared to smaller QDs [94,95,96]. Thus, it further shows the differences in toxicity between types of quantum dots and highlights the need to further investigate QDs and their complexity. On the other hand, both our red CdSe/ZnS QDs (Table 4) and the previously reported green QDs [28,29] caused alteration in many genes related to translation-associated ribosomal components. In red CdSe/ZnS QDs, many altered genes in the large ribosomal subunit, such as *RPL30, RPL25, RPL28,* and *RPL48A* (Table 4), are essential genes that participated in the cytoplasmic translation, and nulled mutants of these genes led to inviable cells in a genetic study by Steffen et al. in 2012 [97]. Additionally, although QDs did not affect the expression of actin dynamic regulatory genes, we found an alteration in their protein expression (Figure 12). This implies that CdSe/ZnS QDs’ ability to modulate translation may also be a cause for their toxicity. It is well known that a number of post-transcription processes could influence protein translation and expression; thus, it is reasonable to hypothesize that QDs could have modulated one of these processes to alter the protein expression. One factor that dictates how much protein could be expressed is the stability of mRNA, as this would determine the amount of mRNA available for translation. Furthermore, some ribosomes specifically chose their mRNA target to translate and, therefore, QDs could have influenced the ribosomal activity to favor the translation of certain proteins. Post-translation-wise, QDs could influence protein life span or directly modify certain proteins by direct interaction. Furthermore, a number of mitochondrial-associated genes were altered by QD treatment (Table 4 and Table 5); thus, QDs could have caused a reduction in energy production and, thus, indirectly influence translation as well as other cellular processes.

## Figures and Tables

**Figure 1 cells-12-00484-f001:**
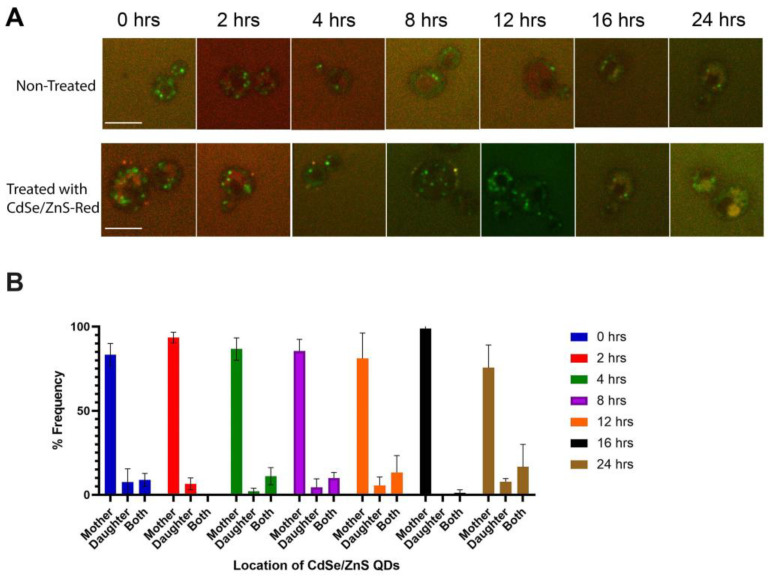
QDs’ external association with yeast. Green signals are expressed Vps10-GFP and red signals are CdSe/ZnS QDs. The size bar is equivalent to 5 μm. (**A**) QDs attach to the surface of yeast cells. (**B**) Percent of cells (% Frequency) that had QDs attached to mother site, daughter/budding site, or at the neck/both mother and daughter site.

**Figure 2 cells-12-00484-f002:**
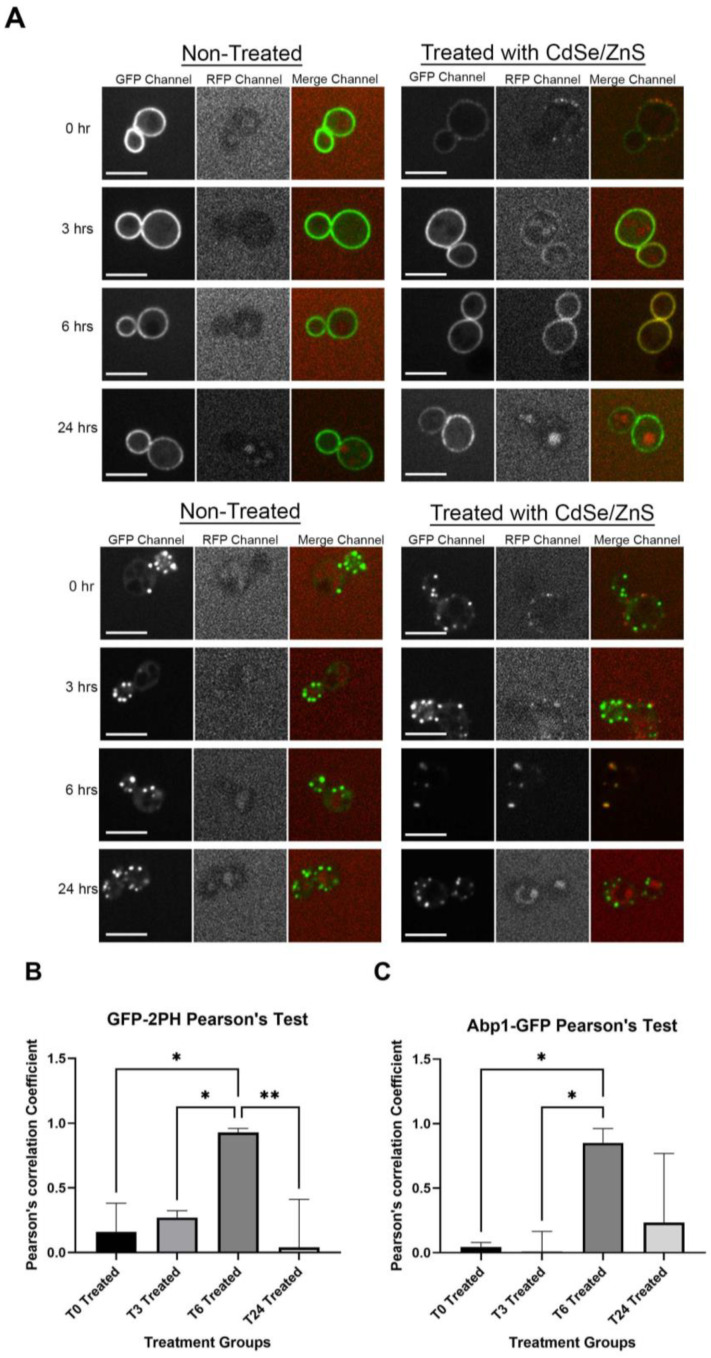
QDs’ intracellular trafficking. Cells expressing GFP-2PH or Abp1-GFP were cultured in SD-HIS for 0 h, 3 h, 6 h, or 24 h with or without QDs (4 μg/mL). The images show different channels (in order: GFP, RFP, and merge) for control on the left and QD-treated on the right. The size bar is equivalent to 5 μm. (**A**) Confocal microscopy images of the plasma membrane reference marker GFP-2PH (top) and confocal image of the early endocytosis reference marker (bottom). Images B and C show the graphs of Pearson’s correlation coefficient of QDs with different reference markers. (**B**) Graph for QDs and plasma membrane colocalization and (**C**) graph for QDs and the early endocytosis vesicle colocalization. * *p* < 0.05, ** *p* < 0.01.

**Figure 3 cells-12-00484-f003:**
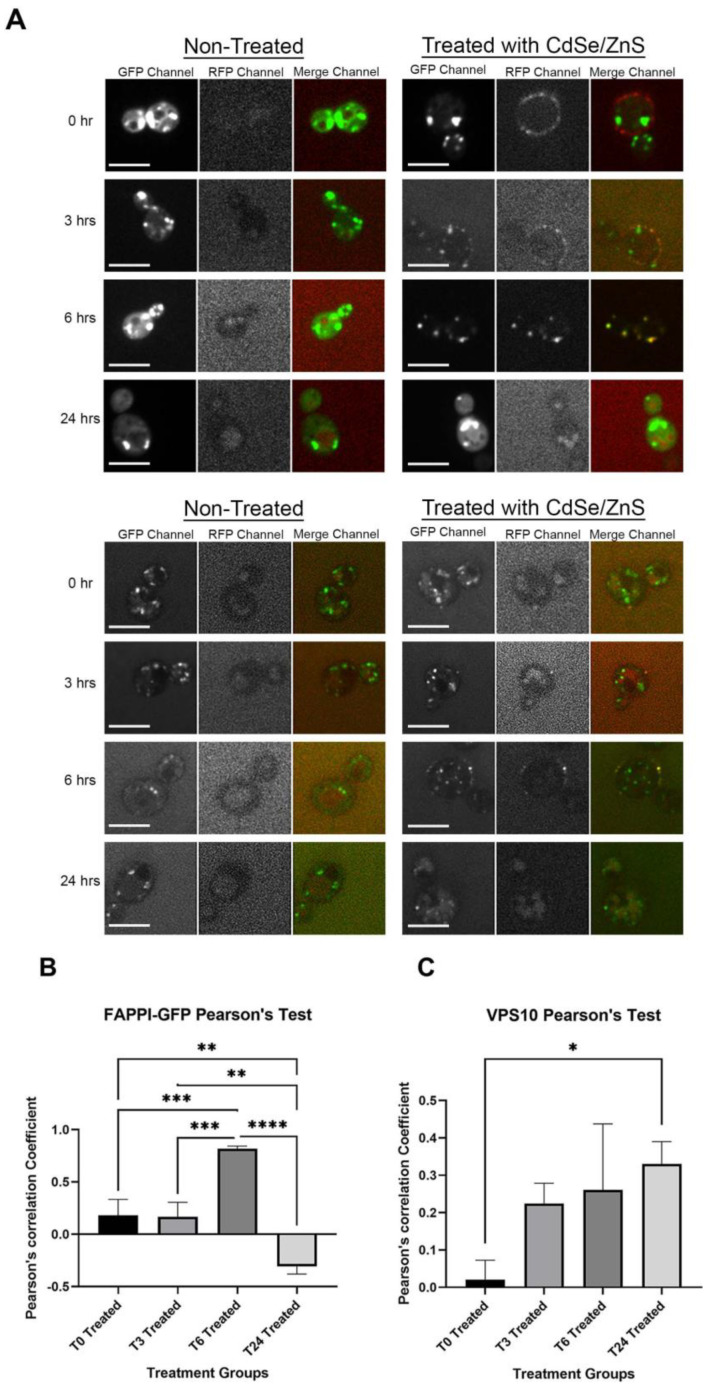
QDs’ intracellular trafficking. Cells expressing FAPPI-GFP or Vps10-GFP were cultured in SD-HIS for 0 h, 3 h, 6 h, and 24 h with or without QDs (4 μg/mL). The microscopic images were obtained from different channels (in order: GFP, RFP, and merge) for control on the left and QD-treated on the right. The size bar is equivalent to 5 μm. (**A**) Confocal microscopy images of the late Golgi/trans-Golgi network (top) and confocal image of the late endosome (bottom). Images (**B**) and (**C**) showed the graphs of Pearson’s correlation coefficient of QDs with different reference markers. (**B**) Graph for QDs and FAPPI-GFP colocalization and (**C**) graph for QDs and the Vps10-GFP colocalization. * *p* < 0.05, ** *p* < 0.01, *** *p* < 0.001, **** *p* < 0.0001.

**Figure 4 cells-12-00484-f004:**
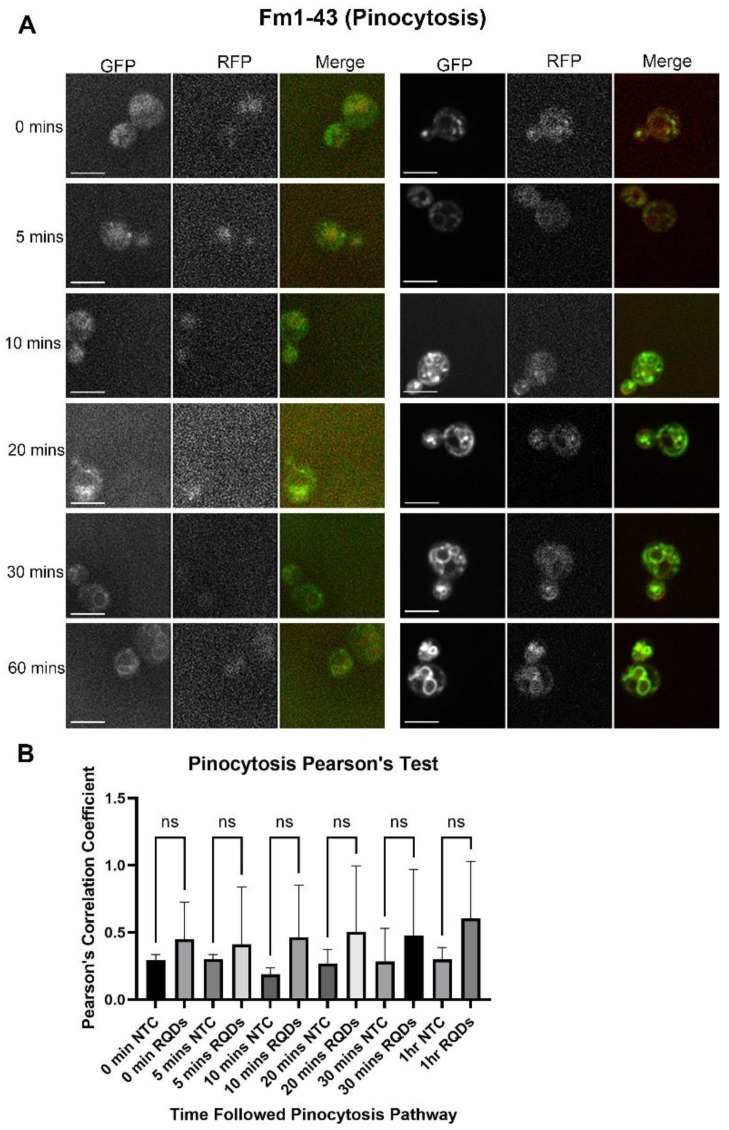
A pulse and chase experiment to determine the degree of colocalization between red quantum dots (RQDs) and green pinocytosis tracking dye Fm1-43. Fm1-43-labeled cells at 0 °C were incubated at room temperature for varying durations up to 60 min. The size bar is equivalent to 5 μm. (**A**) Confocal microscopy of the pinocytosis pathway within 60 min (left—untreated, right—treated) and (**B**) graph of Fm1-43 and QDs colocalization assessed by Pearson’s coefficient.

**Figure 5 cells-12-00484-f005:**
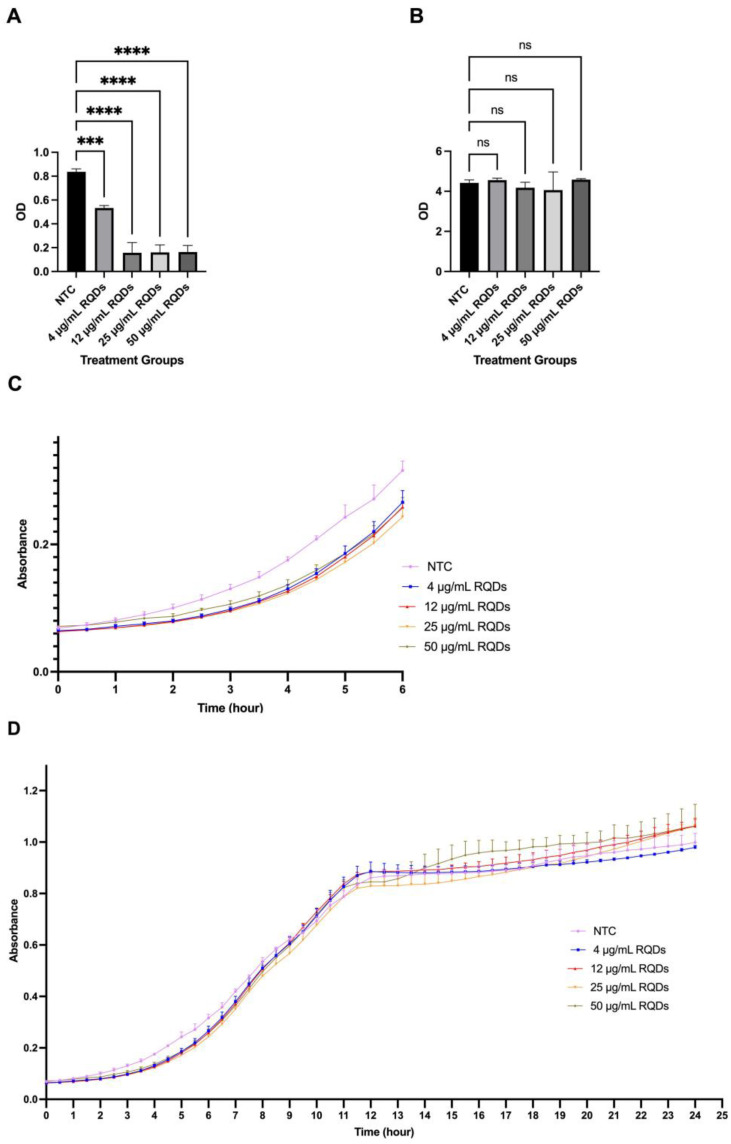
Growth assay to determine the growth rate of yeast in the presence of QDs in SD-HIS. (**A**) Growth assay for yeast cells at 6 h after QD treatment. (**B**) Growth assay for yeast cells at 24 h after QD treatment. (**C**) A 6 h kinetic growth curve of yeast in the presence of quantum dots in various concentrations (0 μL/mL, 4 μg/mL, 12 μg/mL, 25 μg/mL, and 50 μg/mL). (**D**) A 24 h kinetic growth curve of yeast in the presence of quantum dots in various concentrations (0 μL/mL, 4 μg/mL, 12 μg/mL, 25 μg/mL, and 50 μg/mL). *** *p* < 0.001, **** *p* < 0.0001.

**Figure 6 cells-12-00484-f006:**
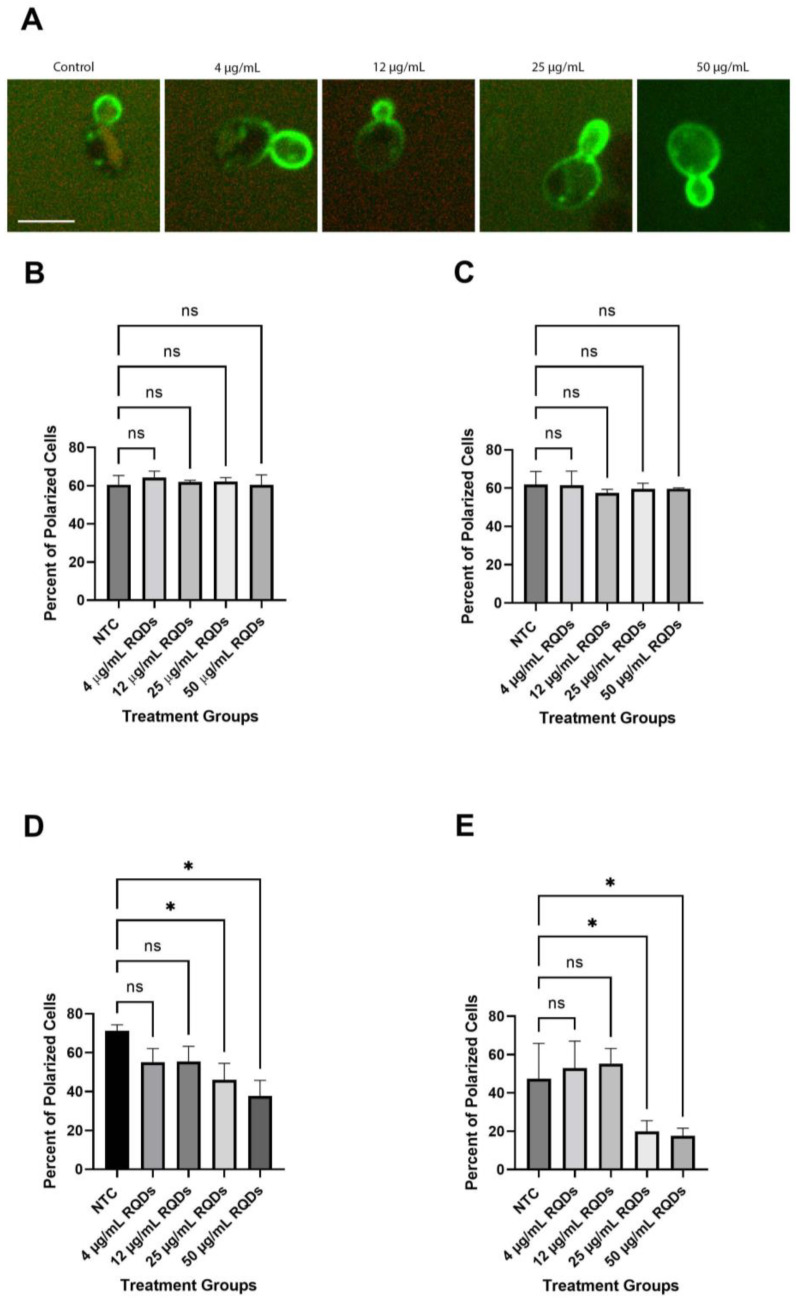
The distribution of GFP-Snc1 on the plasma membrane. For each, 30 cells were counted. The size bar is equivalent to 5 μm. (**A**) Images of cells with different polarization at 6 h post-QD treatment. (**B**) The percentage of cells showed the polarized distribution of GFP-Snc1 on the plasma membrane for 1 h after treatment. (**C**) The percentage of cells showed the polarized distribution of GFP-Snc1 on the plasma membrane 3 h after treatment. (**D**) The percentage of cells showed the polarized distribution of GFP-Snc1 on the plasma membrane at 6 h after treatment. (**E**) The percentage of cells showed the polarized distribution of GFP-Snc1 on the plasma membrane at 24 h after treatment. * *p* < 0.05.

**Figure 7 cells-12-00484-f007:**
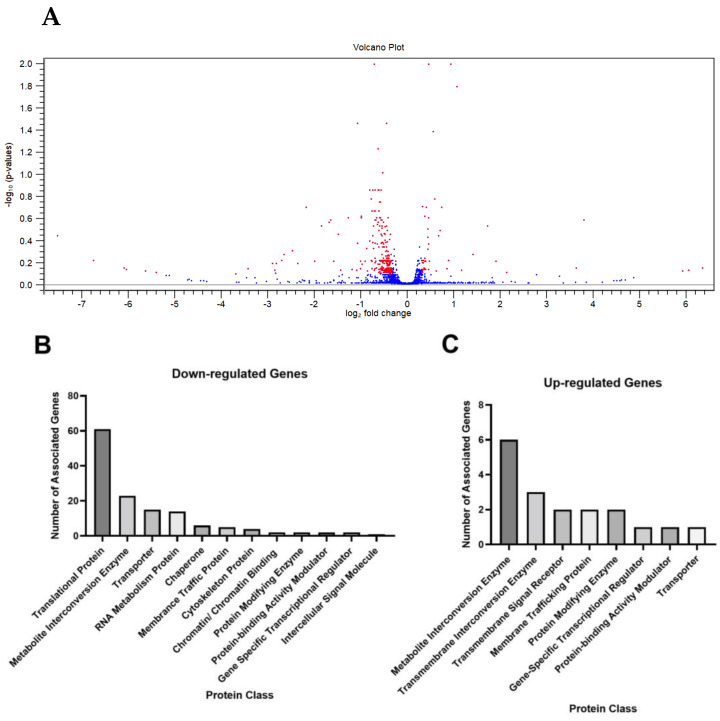
RNAseq analysis. (**A**) A volcano plot showing altered genes in RNA-Seq analysis with red dots indicating genes that have a fold change larger than 1.25 and less than −1.25 with *p*−value < 0.05. (**B**–**E**) Bar graphs represent classes of genes that were differentially expressed due to the treatment of QDs. Each bar is the number of differentially expressed genes in a protein class or associated with a biological process. (**B**) Downregulated genes sorted by protein class. (**C**) Upregulated genes sorted by protein class. (**D**) Downregulated genes are sorted by associated biological processes. (**E**) Upregulated genes sorted by associated biological processes.

**Figure 8 cells-12-00484-f008:**
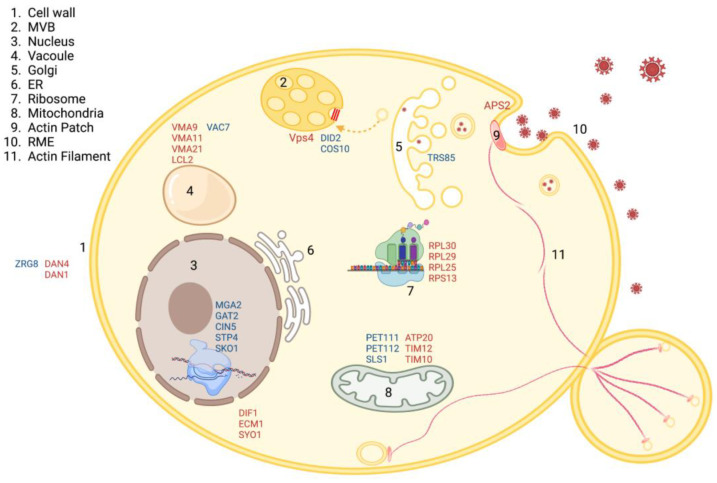
Model of important upregulated genes (blue) and downregulated genes (red) upon CdSe/ZnS QDs treatment, as shown by RNA-seq data. Upregulated genes include genes associated with the mitochondria structure and function (PET111, PET112, and SLS1), genes involved in the vesicular trafficking (DID2, COS10, and TRS85), a gene involved in cell wall integrity (ZRG8), transcription regulation (MGA2), and vacuole structure (VAC7). Important downregulated genes include genes involved in translation (RPL30, RPL29, RPL25, and RPS13), mitochondria and metabolism-related genes (ATP20, TIM12, and TIM10), transcription factor genes (GAT2, CIN5, STP4, and SKO1), cell wall integrity genes (DAN1 and DAN4), endocytosis regulation gene (APS2), and vacuole-associated genes (VMA9, VMA11, VMA21, and LCL2). Created with BioRender.com.

**Figure 9 cells-12-00484-f009:**
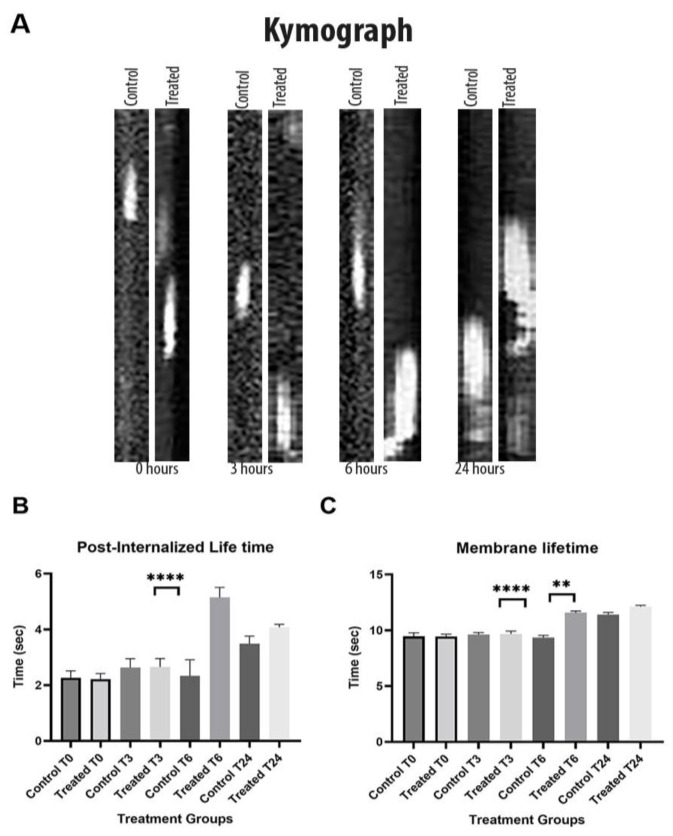
Inefficient endocytosis and prolonged path duration of the endocytosis vesicle resulted from the slowed detachment of Abp1-GFP. (**A**) Kymograph of endocytic vesicle formation showing inefficient endocytosis for groups treated with 25 μg/mL of CdSe/ZnS at 0 h, 3 h, 6 h, and 24 h post-treatment. (**B**) Prolonged lifetime of Abp1-GFP on the plasma membrane. (**C**) Lifetime of Abp1-GFP post-membrane detachment. ** *p* < 0.01, **** *p* < 0.0001.

**Figure 10 cells-12-00484-f010:**
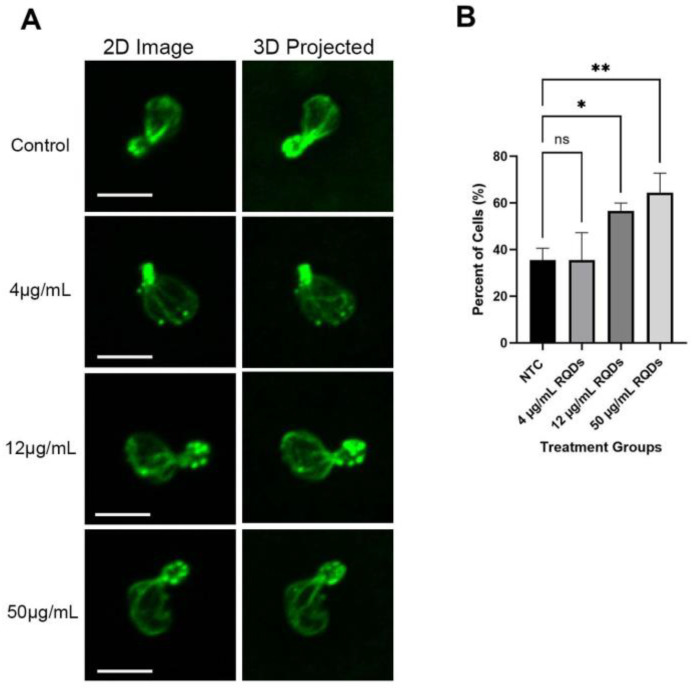
Actin filament integrity assay. (**A**) Cells were treated with an increasing concentration of QDs for 6 h and stained with Actistain 488. Cells were then visualized using the confocal microscope. The left column is a flat, 2D image taken at the mid-plane of cells when cells are in focus. The right columns are the projected pictures from 3D images that allow the visualization of all cables inside yeast cells. Each row signifies images from a treatment group, in the top-down order: control, 4 μg/mL, 12 μg/mL, 25 μg/mL, and 50 μg/mL. The size bar is equivalent to 5 μm. (**B**) The percentage of cells with abnormal actin cable was quantified and graphed. A total of 30 cells were used for each sample. * *p* < 0.05, ** *p* < 0.01.

**Figure 11 cells-12-00484-f011:**
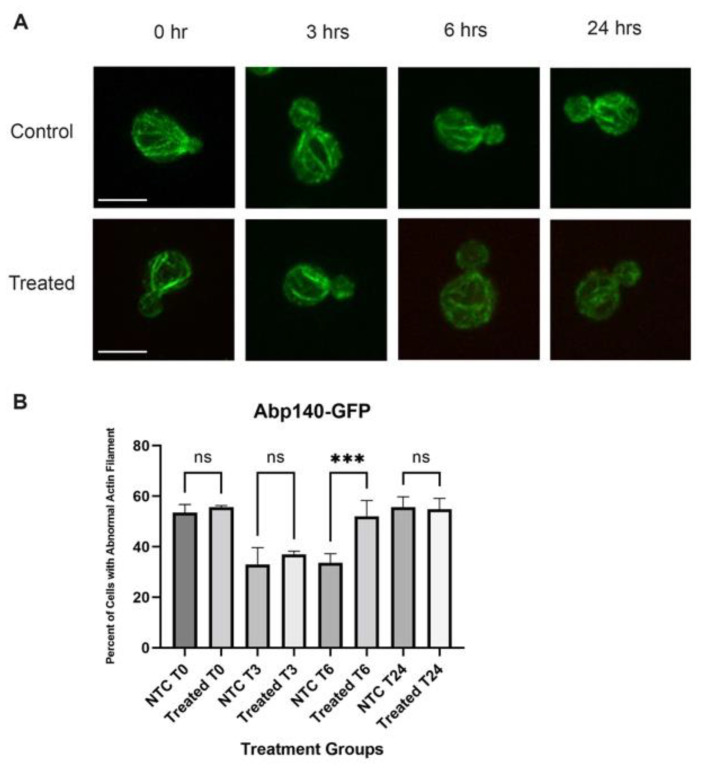
Actin cable integrity assay. (**A**) Abp140-GFP-expressing cells were treated with QDs and visualized over time using the confocal microscope. The size bar is equivalent to 5 μm. (**B**) Actin cables of 30 cells were assessed for their structure. *** *p* < 0.001.

**Figure 12 cells-12-00484-f012:**
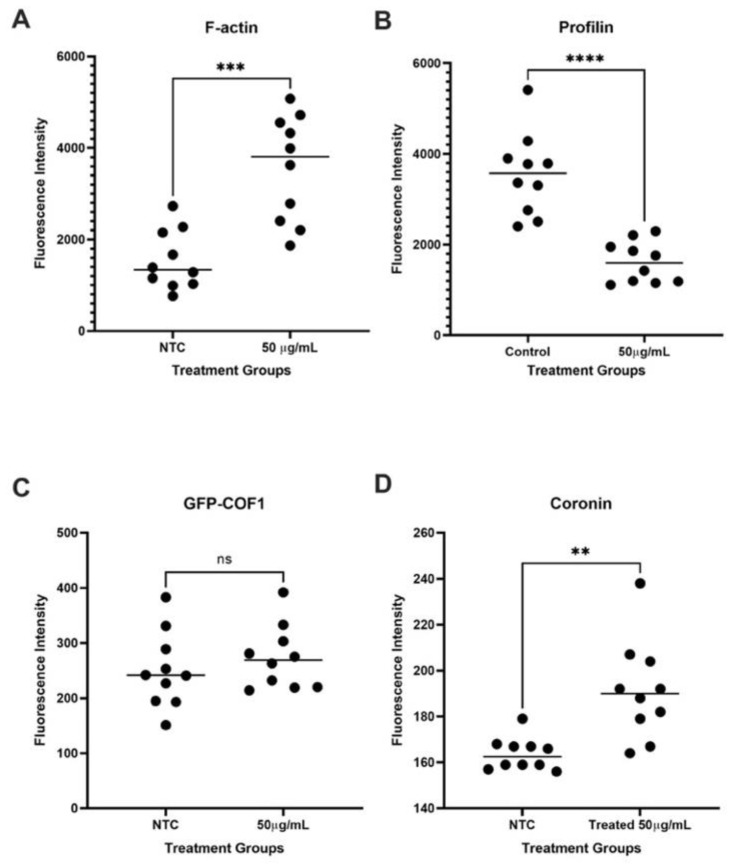
Graphs show the level of GFP-tagged actin-associated proteins. (**A**) F-actin stained with Actistain488 dye, (**B**) Pfy1-GFP, (**C**) GFP-Cof1, and (**D**) Crn1-GFP. ** *p* < 0.01, *** *p* < 0.001, **** *p* < 0.0001.

**Figure 13 cells-12-00484-f013:**
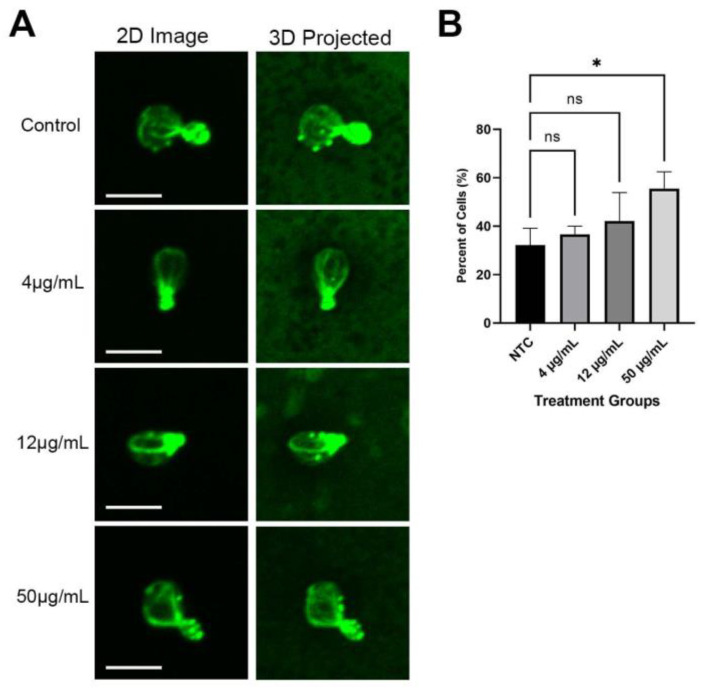
A recovery assay was performed where cells exposed to QDs were moved to a QD-free media for 3 h and the morphology of the actin cable was assessed. (**A**) Images of the actin cable when stained with Actistain 488 and visualized with the confocal microscope. The left column is a flat, 2D image taken at the mid-plane of cells when cells are in focus. The right column is the projected pictures from 3D images that allow the visualization of all cables inside yeast cells. Each row signifies images from a treatment group, in the top-down order: control, 4 μg/mL, 12 μg/mL, 25 μg/mL, and 50 μg/mL. The size bar is equivalent to 5 μm. (**B**) The percentage of cells with abnormal actin cable was quantified and graphed. A total of 30 cells were used for each sample. * *p* < 0.05.

**Figure 14 cells-12-00484-f014:**
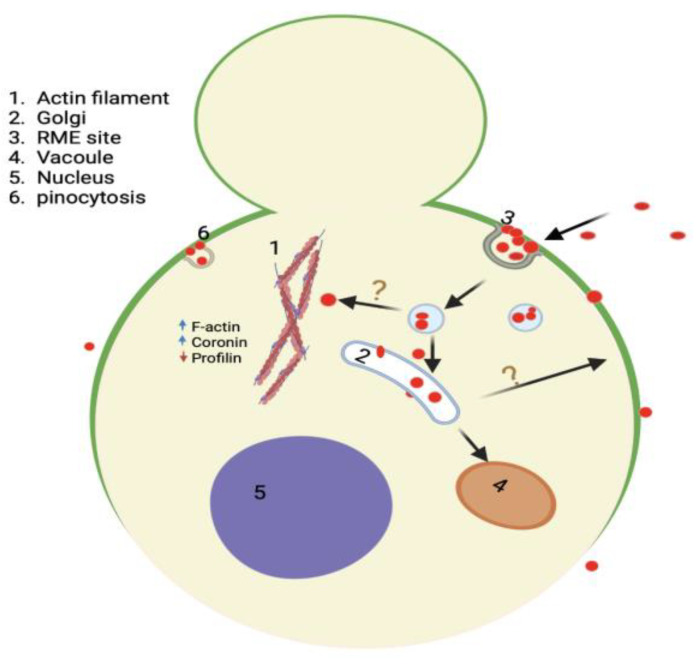
Model summarizing the interaction and impact of QDs in yeast, including their mode of entry mainly by endocytosis, and their trafficking to the Golgi. The model also shows multiple impacts QDs have on yeast cells, such as loss of polarized GFP-Snc1 membrane distribution and alteration of actin dynamic regulatory protein to support actin polymerization and bundling. Created with BioRender.com.

**Table 1 cells-12-00484-t001:** Yeast strains used in this study.

Strain	Status	Media	Genotype
S288C	Existing	YPD	*MATα SUC2 mal met gal2 CUP1 flo1 flo8-1 hap1*
Abp1-GFP	Existing	SD-HIS	*MATα ABP1-GFP-HIS3 his3∆1 leu2∆ met15∆ ura3∆*
GFP-Snc1	Existing	SD-URA	*MATα his3∆1 leu2∆ met15∆ ura3∆ SNC1-GFP-URA*
GFP-2PH	Existing	SD-URA	*MATα his3∆ura∆leu∆trp∆lys∆ GFP-2PH (PLC)-URA3*
Vps10-GFP	Existing	SD-HIS	*MATα his3∆1 leu2∆ met15∆ ura3∆ pRS426 GFP-PH FAPPI*
Abp140-GFP	This study	SD-HIS	*MATα his3∆1 leu2∆0 met15∆0 ura3∆0 Abp140-GFP-HIS*
GFP-Cof1	This study	SD-URA	*MATα his3∆1 leu2∆0 met15∆0 ura3∆0 GFP-Cof1-URA*

**Table 2 cells-12-00484-t002:** Mean size and zeta potential of QDs in water, nutrient-lacking histidine media (−His +Glu), and yeast peptone dextrose media (YPD) using DLS. The polydispersity index (PDI) value is also provided. PDI that is less than 0.1 represents a highly monodisperse sample, usually a clean standard. In contrast, a PDI value greater than 0.7 indicates that the sample has a very broad size distribution and may have agglomerations.

	Sample	QD + Water		QD + HIS		QD + YPD	
		Mean Intensity	PDI	Mean Intensity	PDI	Mean Intensity	PDI
Size (nm)	Run-1	17.2	0.168	1372.5	0.994	178.0	0.309
	Run-2	15.6	0.106	906.9	0.882	169.7	0.122
	Run-3	17.6	0.061	2091.2	0.856	184.1	0.240
	Avg	16.8	0.112	1456.9	0.911	177.3	0.224
ZP (mV)	Run-1	−49.2		3.4		0.7	
	Run-2	−49.5		3.2		0.6	
	Run-3	−49		3.3		0.9	
	Avg	−49.23		3.30		0.73	

**Table 3 cells-12-00484-t003:** Agglomeration assay. QDs incubated in water with a pH ranging from 1 to 7 and incubated for 24 h. The samples were spun to isolate largely agglomerated QDs at the bottom; the supernatant was then tested using ICP-OES.

	Average Cd Content (ppm)
Control + QDs	2.038
pH 1 + QDs	0.977
pH 2 + QDs	1.926
pH 3 + QDs	2.032
pH 4 + QDs	1.920
pH 5 + QDs	2.111
pH 6 + QDs	2.112
pH 7 + QDs	2.093

**Table 4 cells-12-00484-t004:** Agglomeration assay. QDs in SD-HIS and YPD with natural pH (4.86 and 5.57) and neutral pH (pH 7). The samples were spun to isolate largely agglomerated QDs at the bottom; the supernatant was then tested using ICP-OES.

	Average Cd Content (ppm)
YPD pH 5.57	1.049
YPD pH 7	1.107
−His + Glu pH 4.86	0.852
−His + Glu pH 7	1.069

**Table 5 cells-12-00484-t005:** Upregulated genes that showed the impact of red CdSe/ZnS QDs on membrane trafficking, metabolism, and cellular processes. Genes are sorted by protein class.

Upregulated Genes	
Vesicular Trafficking (6)	*DID2, COS10, VAC7, ECM21, TRS85, SND1*
Zinc Ion Homeostasis (3)	*IZH4, IZH1, IZH2*
Transcription Factors (5)	*GAT2, CIN5, HAP4, SKO1, STP4*
Enzymes and Protein Kinases (10)	*TOS3, MPS1, BIO2, ENA2, ARO10, NFL1, CIP1, DPB2, SAP1, AFT1*
Mitochondria (3)	*PET111, PET112, SLS1*
Cell Wall Organization (1)	*ZRG8*
Eisosome (2)	*SEG2, SEG1*
Fatty Acid Metabolism (10)	*OLE1, FAA4, MGA2, PBL2, ICT1, NEM1, TDA4, TDA4, HMG2, OSH2, SCT1*

**Table 6 cells-12-00484-t006:** A list of downregulated genes showed the impact of red CdSe/ZnS QDs on membrane trafficking, cell wall organization, cytoskeleton, metabolism, and protein translation. Genes are sorted by protein class.

Downregulated Genes	
Ribosomal large subunits (28)	*RPL46A, 37A, 26B, 23B, 31A, 37B, 22B, 38, 43B, 33B, 22B, 43A, 33A, 39, 30, 29, 25, 42B, 35A, 40A, 14A, 24A, 35B, 28, 42A, 34B, 7B, 14B*
Ribosomal small subunits (18)	*RPS28A, 29A, 29B, 23B, 28B, 30B, 27B, 10B, 17A, 13, 25A, 27A, 25B, 31, 21A, 19A, 21B, 22A*
Ribosome associated (7)	*RPS30A, TMA7, TMA10, TMA16, YET2, MRT4, FAP7*
Mitochondria function (32)	*ATP20, ATP18, ATP19, ATP15, TIM8, TIM12, TIM10, TIM11, COA2, FMP45, TMH18, RSM19, MIX14, MRPL50, MRPL37, MRPS16, MTC3, EMI1, CMC1, BOL1, COA4, MIC12, MRP10, MRP2, COX7, FIS1, MRPS12, MRPL51, TRX1, HSP10, OAC1, QCR8*
RNA processing (13)	*LSM3, NOP10, RDS3, IST3, HUB1, YSF3, LSM7, DIB1, POP5, POP8, LSM6, APQ12, SMX3,*
Stress response (21)	*ROQ1, GRX1, HOR7, ATX1, MHF1, MHF2, SRX1, TMA10, GRE1, DDR2, POP8, HMF1, HSP12, SPR28, MEI5, SPO19, PAU20, PAU19, AUS1, FMP16, SDD2*
Cell wall integrity (4)	*DAN4, DAN1, FMP45, TIR4*
Sporulation and meiosis (7)	*BNS1, SHU2, EMI1, SPR28, MEI5, SPO19, FMP45*
Protein folding/chaperone/Degradation (7)	*PFD1, GIM3, ROQ1, RUB1, HSP10, POC4, EMC6*
Transporter (10)	*PHO89, AUS1, PMP3, SIT1, OAC1, DRE7, HXT6, VMA11, VMA9, VMA21*
Vacuole (4)	*VMA9, VMA11, VMA21, LCL2*
Vesicular trafficking and Secretion (21)	*YOS1, TRX1, SNX3, APS2, EFM4, MLC2, DYN2, APR10, TCA17, LDB18, SFT1/NCE101, YSY6, SBH2, SRT1, GOT1, SSA3, KSH1, SPC2, SSS1, OST1*
Transcription (13)	*MBF1, TFB5, FYV5, HMRA1, SPT4, HHT1, AHC2, RPB11, SRB6. RPC11, RPO26, RPC10, YFR1-2*
Nuclear import and export (3)	*DIF1, ECM1, SYO1*
Membrane lipid synthesis and homeostasis (6)	*CHO1, OST1, SRT1, TSC3, OPI11, OPI3*
Enzymes (20)	*FCY1, CTF8, TSC3, RNH203, PBI2, OPI1, VMA9, VMA21, WIP1, BNA1, BNA2, SSA3, REE1, HMRA1, URA10, RPC10, POC4, CFD1, HIS6, DFR1*
Replication (1)	*CTF8*
Apoptosis (1)	*SDD2*

## Data Availability

The data are contained within the article.

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
