# Peer review of "Red CdSe/ZnS QDs’ Intracellular Trafficking and Its Impact on Yeast Polarization and Actin Filament"

_cells, 2023, doi:10.3390/cells12030484_

Round 1

Reviewer 1 Report

In this study, Le et al. investigated the biological impact of CdSe/ZnS quantum dots (QDs) on the yeast Saccharomyces cerevisiae. They monitored the internalization of QDs into the yeast to study their endocytosis mechanism and subcellular localization, and figured out QDs affect the actin dynamics via RNA sequencing and further mechanism studies. This work explores toxicity mechanism of CdSe/ZnS QDs on yeast and offers important knowledge for further QD applications. However, there are several issues that should be resolved before publication as noted below.

1. RFP channels in Figures 2-4 should exhibit the same trend, independent of the GFP strain, but they appear strain-dependent. Please explain the reason and confirm the RFP channel images are not affected by false signals such as strong background or spillover from the GFP channel.

2. Actastain 555 is likely to have overlapping excitation and emission with red QDs, which were used with 561 nm excitation and showed 620-635 nm emission. Actin staining should be done with dyes with different excitation/emission wavelengths such as far-red emission dyes.

3. Polydispersity (PDI) results are recommended to be provided for the DLS results in Table 2.

4. Statistical analyses carried out in Figures 2B, 3B, and 4B might get more meaningful if the comparison can be done with different control groups. The comparison with non-treated groups cannot reveal any information regarding the co-localization of GFP and RFP signals in QD-treated groups since non-treated groups have no meaningful signal in the RFP channel. For example, the analysis between different time points would be more insightful.

5. In Figure 6A, it is recommended to provide low-magnification images showing multiple cells.

6. Please explain the relationship between adaptor protein 2 and genes listed in Table 6.

7. All microscopy images need to have scale bars.

8. In the caption of Figure 1a, it is recommended to explain what each color in confocal images stands for.

9. Before using abbreviations (such as Abp, NTC, RQD, etc), they should be defined in the text. Especially, ABP is one of the key elements in this work, but the term can indicate a few different proteins, such as auxin binding protein and actin-binding protein, so it should be properly defined.

10. Section 3.1 is repeated twice. Corrections should be made to the section numbers.

11. Please check if the subfigure numbers in Figure 7 are correct in the caption and main text.

Author Response

Please find the attached rebuttal letter. 

thanks.

Reviewer 2 Report

In this article, Li et al. evaluated CdSe/ZnS QDs the trafficking path in Saccharomyces cerevisiae and its impact on yeast intracelular dynamics. The article is relevant with an interesting in vitro approach. However, several aspect need attention to validate the obtained results:

1. In QDs Agglomeration results, please indicate polidispersity index which also can be used to compared agglomeration behavior amongs samples.

2. In figure 1A, the third row of images is not described. Please clarify the meaning of those images.

3. One of my biggest concern is the setting of adquisition parameters in the confocal images of QDs’ Subcellular Trafficking experiments. As can be seen in the RFP channels images. The signal to noise ratio is very low and not equal between the different panels. Therefore it is difficult to differentiate the true signal from the QDs. In some not treated images it could be observed red dot fluorescence due to autoflorescence? This has an negative impact in the Pearson coefficient determination and the further comparision.
4. Please add the scale bars to the images.

5. Authors claim that “Our confocal microscopic images for non-treated cells revealed that QDs partially co-localize with the pinocytosis reference dye”. How is that possible if not QDs were used? This images are the most affected by autofluorescence and a poor signal to noise ratio.

6. The assumption provided by authors related to grown inhibition at 6 h and not at 24 h needs experimental validation. A kinetics of the growth curve at different times could help to clarify the differences observed at the reported times. 
7. Why were the following assays  performed at such high concentrations of QDs? If growth inhibition changes are observed at lower concentrations? Plase clarify because the changes could be due to cytotoxic effects rather than inhibitory effects.

Author Response

Please find the rebuttal letter attached.

Thanks.

Round 2

Reviewer 1 Report

The authors addressed a major part of the previous comments so it is recommended to accept it as it is.

Reviewer 2 Report

The authors have responded to my comments and made the pertinent changes. I have no more comments.